# Bifrontal Osteoplastic Flap: An Option to Decrease Infection in Bifrontal Craniotomies with Skull Base Osteotomies

**DOI:** 10.3390/brainsci12020163

**Published:** 2022-01-26

**Authors:** Michael Ortiz Torres, Endrit Ziu, Samiat Agunbiade, Steven B. Carr, N. Scott Litofsky

**Affiliations:** Division of Neurosurgery, School of Medicine, University of Missouri, Columbia, MO 65212, USA; ortiztorresm@health.missouri.edu (M.O.T.); ziue@health.missouri.edu (E.Z.); samiat.agunbiade@gmail.com (S.A.); carrbs@health.missouri.edu (S.B.C.)

**Keywords:** bifrontal craniotomy, skull base, meningioma, esthesioneuroblastomas, infection, osteoplastic flap

## Abstract

Infection can be a common complication following bifrontal craniotomy with skull base osteotomies given the potential violation of sinuses and entry into the nasal structures. Our objective was to examine our series of patients who underwent a bifrontal craniotomy with skull base osteotomies and describe the infection rate. We propose the bifrontal osteoplastic flap as an adjunct to infection prevention. A retrospective single-center study of a patient database was performed. Twenty patients were identified. Fifty-five percent were male. The mean age was 55.7 ± 13.9 years. The most common indications for surgery were esthesioneuroblastomas (35%) and anterior skull base meningiomas (30%). Six patients (30%) developed an infection, 1 patient (5%) developed a CSF leak, and no patients developed a mucocele. All 6 infected cases had nasal pathology with intracranial extension, they all received chemoradiation post-operatively and were all combined cases with otorhinolaryngology. Eighty-three percent of these patients required a craniectomy and all of them required long-term IV antibiotics. Infection is not uncommon after a bifrontal craniotomy with skull base osteotomies and the use of the bifrontal osteoplastic flap in cases where the risk of infection is high, i.e., esthesioneuroblastomas surgery, may help reduce said risk and lead to better patient outcomes.

## 1. Introduction

Lesions of the anterior cranial fossa floor, including benign and malignant primary or secondary tumors, cerebrospinal fluid fistulas, encephaloceles and other skull base defects, are often approached through a bifrontal craniotomy, with or without a skull base osteotomy, such as a cribriform osteotomy. The exposure provides a generous corridor for the management of such midline lesions, regardless of size [1]. Usually, the frontal sinuses are entered, which creates a clean contaminated surgical field. Likewise, with cribriform osteotomies, the ethmoid sinuses are exposed, which also contributes to the contaminated field. While clean, the contaminated field increases the risk of infection [2,3]. When a durotomy at the cribriform plate is performed, a cerebrospinal fluid (CSF) leak can occur, which leads to the potential for developing a post-operative mucocele or an infection [4].

When a bifrontal craniotomy is complicated by infection, the bone flap may need to be removed and the patient may require treatment with intravenous antibiotics for an extended period of time [5,6]. Removal of the bone flap creates a significant cosmetic deformity, and the loss of skull integrity reduces brain protection. To solve both problems, cranioplasty after the completion of antibiotic treatment is necessary; additional associated risks ensue, infection being the most predominant [7,8]. Prevention of infection from the initial surgery helps avoid these significant morbidities associated with this complication.

We reviewed our series of patients who underwent bifrontal craniotomies with skull base osteotomies for skull base lesions or skull base repair with the aim of describing the overall rate of complications, particularly infections. After consideration of these outcomes, we propose an operative solution to reduce the risk of infection, which we now employ.

## 2. Materials and Methods

This retrospective single-center study was carried out in a tertiary care hospital and was approved by the University of Missouri School of Medicine Institutional Review Board (IRB) (IRB#2022367) and conducted in compliance with Health Insurance Portability and Accountability Act regulations, with waiver of patient consent as the study was retrospective and involved deidentified patient data

The University of Missouri Neurological Surgery patient database was queried to identify all patients who had undergone bifrontal craniotomies from August 2006 through January 2020. Patients who had a bifrontal craniotomy for skull base tumors or repairs were included. Cribriform osteotomies, whether transcranial or transnasal, were performed in all cases. Craniectomies for traumatic injuries were excluded. Data were extracted from the patients’ electronic medical records. These data included inpatient and available outpatient follow-up visits. Recovered variables included gender, age at diagnosis, indication for surgery, clinical presentation, type of skull base repair, complications and need for adjuvant treatment, among others. Descriptive statistics were performed with Excel 2016 (Microsoft, Redmond, WA, USA). Continuous variables were summarized as the mean ± standard deviation.

After describing our series, we discuss our experience with the bifrontal osteoplastic flap as an option for decreasing the infection rate. The technical nuances of the technique are discussed in detail and our preliminary findings are presented.

## 3. Results

### 3.1. Patient Characteristics

Twenty patients who underwent a bifrontal craniotomy qualified for inclusion. The mean age was 55.7 ± 13.9 years. Fifty-five percent of patients were male. Indications for surgery included esthesioneuroblastomas (35%), anterior cranial fossa meningiomas (30%), nasal carcinomas (25%) and cribriform defects (10%), as seen in Table 1. Only one patient had pre-operative chemoradiation due to recurrent sinonasal adenocarcinoma, which had been treated 30 years prior. All patients who received chemotherapy or radiation had these additional interventions administered 6 weeks after surgery. All joint cases with otorhinolaryngology (13 cases) consisted of a transcranial approach in combination with a transnasal approach. Nasal flaps were not applied, as these cases required extensive debridement and resection of the nasal mucosa and the remaining tissue was not sufficient to repair the skull base defect.

### 3.2. Complications

Six patients (30%) developed an infectious complication (Table 2). No mucoceles were observed and only one patient suffered a CSF leak. All 6 patients with infection had nasal pathology with intracranial extension. They all received post-operative chemoradiation, and all were combined cases with otorhinolaryngology. Post-operative chemotherapy was associated with a statistically significant risk of post-operative infection. Mean operative time for patients with an infection (534.2 ± 77 min) did not differ from those who did not have an infection (453.6 ± 122.3 min, *p* = 0.16). Three patients (50%) had involvement of their skull base repair as part of their infectious process, and all had their grafts removed as part of their debridement craniectomies to treat the subsequent infections. All reconstructive allografts, irrespective of whether or not they appeared to be infected, were removed during the revision. Autografts were only removed if clearly infected. A temporalis fascia graft and/or fat graft was used to seal the defect and separate cranial and sinonasal cavities. The most common organism cultured was methicillin-resistant *S. aureus* (MRSA), present in 4 cases (66.7%). Five (83.3%) of the patients required a craniectomy to treat the infection. Four patients received a polyether ether ketone (PEEK) cranioplasty for reconstruction after treatment of the infection. The mean time to cranioplasty was 22 ± 6 weeks. No patient suffered intracranial hypotension or any other complication related to delayed cranioplasty. One patient who did not undergo post-treatment cranioplasty wished to avoid additional surgery.

## 4. Discussion

### 4.1. Infection Rate for Bifrontal Craniotomies

Bifrontal craniotomies with skull base osteotomies provide operative access to lesions of the anterior cranial fossa and permit adequate skull base reconstruction. The sinonasal structures are usually exposed. These procedures have an increased risk of infection compared to other craniotomies [5]. The rate of bone-flap infection in all types of craniotomies varies from 1–11% depending on factors such as the type of surgery, duration of surgery greater than 4 h, occurrence of CSF leak, and prior radiation treatments [5,6,9]. The reported rate of infection for bifrontal craniotomies with or without skull base osteotomies is between 1–7%, which is higher than that of all other craniotomy types [2,10,11,12]. Roughly half of these infections correspond to bone-flap osteomyelitis. Notably, these studies report infection rates for a variety of intracranial pathologies with or without nasal cavity and/or sinus involvement approached via a bifrontal craniotomy with or without skull base osteotomies. Although originally reported in the context of decompressive craniectomies, a defect area >125 cm^2^, as well as the presence of a post-cranioplasty fluid collection have been associated with bone flap failure [13,14]. The relatively larger size of bifrontal craniotomies could also represent a risk factor for bone flap infection. Following anatomical knowledge during surgical procedures can reduce the risk of complications, e.g., as a result of tissue traumatization or ischemia [15]. Appropriate handling of tissue can also help.

Reports of infection rates for nasal-originating pathology are very sparse. Palejwala et al. [16] reported a bone flap osteomyelitis rate of 75% for patients operated on who harbored advanced (Stage C or D) Kadish stage esthesioneuroblastomas. Therefore, although the infection rate in our series may appear high at first glance, the rarity of the pathology, as well as the scant amount of data published on the subject may underestimate the complication/infection rate. We are unaware of any other case series of infections in bifrontal craniotomies with skull base osteotomies.

### 4.2. Risk Factors for Infection

Chemotherapy and/or radiation therapy may be significant factors contributing to the development of post-operative infections in this group of patients. All six patients with post-operative wound infections in our series had undergone radiation and chemotherapy treatment. Radiation impairs wound healing, likely increasing the risks of post-operative wound infection [5,9,17]. The mechanism of damage is related to stromal and vascular fibrosis, decreasing microcirculation, causing delayed mucosal healing and creating a nidus for infection [18]. Additionally, chemotherapy agents can cause myeloid suppression, leading to anemia, neutropenia and thrombocytopenia, increasing the risk of infection [18]. Furthermore, none of the patients who underwent bifrontal craniotomy for diagnoses unrelated to malignancies had post-operative infections. All of the infected cases were combined cases with otorhinolaryngology. In these cases, contamination of a clean surgical field is more likely given that there is significant surgical manipulation of the sinonasal structures, sometimes both from the nasal approach and the intracranial approach. The risk of transferring bacteria from the sinonasal cavity to the intracranial space can be reduced by avoiding the exchange of instruments between the surgical teams, but invariably some cross-contamination can occur. The risk of infection with combined otorhinolaryngology and neurosurgery procedures ranges from 0–8.7% [10,19,20]. Having a dedicated surgical technician for each team with its own dedicated set of instruments might be resource consumptive but may help mitigate this risk.

It is well known that the presence of foreign bodies in the intracranial space represents a risk of infection and/or preservation of infection given that these implants are not vascularized and can be colonized with contaminant bacteria [21,22,23]. Only three of our infected cases had allografts as part of the skull base reconstruction and all were removed as part of the craniectomy procedure. We currently routinely use the MEDPOR (Stryker, Inc., Kalamazoo, MI, USA) porous polyethylene implant as it provides rigid support to the reconstruction and allows for rapid ingrowth of blood vessels and soft tissue and subsequent incorporation into bone, which helps strengthen the repair and decrease infection risk [24,25,26]. In our experience using MEDPOR for tegmen defect reconstructions, these implants do not always need to be removed, as its biocompatible nature permits excellent healing with adequate antibiotic therapy [27]. This finding has been supported by other studies and may make MEDPOR a superior option when utilizing allografts for skull base reconstruction [24,28]. However, since no definitive literature supports not removing MEDPOR from anterior skull base reconstructions, where the risk of infection is higher, we removed all allografts in this series. Other factors that may affect the risk of post-operative infection, including comorbidities, type of anesthetic used, curved vs. linear incision, and amount of hair shaved, were not factored into the analysis of this case series [29]. All patients underwent the same perioperative antibiotic protocol consisting of weight-dosed cefazolin (1 g for <80 kg, 2 g for 80–100 kg and 3 g for >100 kg) administered 30 min prior to incision. If the patient was allergic to cephalosporins, 15 mg/kg of vancomycin was administered 2 h prior to incision instead. Antibiotics were continued for 24 h post-operatively. No deviations from protocol occurred in any of the patients included in this study.

### 4.3. A Potential Solution: The Bifrontal Osteoplastic Flap

Based on our concerns related to infections, we decided to perform bifrontal osteoplastic flap craniotomies for our bifrontal exposure in cases where we anticipate violating the frontal sinus or performing anterior cranial fossa osteotomies. In this method, first described by Colohan et al. [21], the anterior portion of the temporalis muscle remains attached to bone bilaterally, creating a hinged vascularized bone flap. By keeping the muscle attached, bone flap vascularization is partially preserved, and the risk of infection is theoretically diminished. We have used this technique on our two most recent patients after analyzing this series.

Our technique for performing the bifrontal osteoplastic flap is illustrated in Figure 1. A standard bicoronal incision is performed, followed by subgaleal dissection and harvest of a vascularized pericranial graft. The pericranial graft is harvested as large as possible, limited laterally by the temporalis muscle bilaterally, superiorly about 1–2 cm posterior to the posterior scalp flap, and anteriorly down to the superior orbital rim. At this point, the midline is identified based on the location of the sagittal suture, and 6 burr holes are marked (Figure 1A). The first two burr holes are marked on the right side about 2 cm lateral to the midline. The anterior burr hole is marked just posterior to the superior orbital rim and the posterior burr hole is marked just anterior to the coronal suture. By keeping the burr holes 2 cm lateral to the midline, we minimize the chances of inadvertent injury to the superior sagittal sinus during drilling. Two burr holes are then marked on each temporalis muscle, the anterior one at the keyhole and the posterior one at a point where a 90-degree angle is created between a line that connects this burr hole to the keyhole burr hole and another line that connects it to the posterior paramedian burr hole. Following this, monopolar electrocautery is used to cut the temporalis muscle areas anteriorly and posteriorly where the burr holes are marked (Figure 1B) bilaterally. The burr holes are then drilled and connected first on the right side in order to elevate the right-sided osteoplastic flap (Figure 1C). The temporalis muscle must be undermined posterior to the keyhole burr hole and anterior to the posterior temporalis burr hole so that the osteotome can be angled superiorly while connecting these two burr holes in order to extend the amount of bone that can be drilled under the muscle. If the bone cuts cannot be fully connected under the temporalis muscle, the bone flap can be epidurally dissected from medial to lateral and then fractured at its base under the temporalis muscle. At this point, the superior sagittal sinus can be epidurally dissected from lateral to medial and the same process is repeated on the left side to elevate the left-sided osteoplastic flap, completing the bifrontal osteoplastic flap (Figure 1D). Avoidance of damaging the temporalis muscle with the drill is important, as is avoiding excessive subperiosteal dissection while performing the above-mentioned steps, as this would render the osteoplastic flap ineffective. For reconstruction, we prefer one small dog bone just medial to each keyhole burr hole, one small dog bone to connect the bone flaps and a burr hole cover at the most posterior paramedian burr hole (Figure 2). Additional hardware may be used as needed for a flush and even fit of the bone flap to prevent cosmetic deformity.

Both of our patients operated on with this technique showed excellent wound healing, and no signs of infection were observed with follow-up of at least 12 months. Both of these patients have similar characteristics to those reported in this series.

In a similar manner, we have revised our previously described middle fossa approach technique [27] to include an osteoplastic flap and have not had an infection after treatment of 12 patients with different pathologies, including vestibular schwannomas, tegmental defects and semicircular canal dehiscence (unpublished data). Subjectively, patients appear to heal faster and more effectively when using this technique. We strongly believe that the additional labor and minimally increased OR times are largely outweighed by the benefits of a vascularized bone flap, and we have not found that our operative field is limited by leaving the flaps attached.

## 5. Conclusions

Infection after bifrontal craniotomy with skull base osteotomies in which the frontal sinus and/or the nasal cavity is entered is not uncommon. Bilateral osteoplastic frontal craniotomies may reduce the risk of infection and lead to better patient outcomes.

## Figures and Tables

**Figure 1 brainsci-12-00163-f001:**
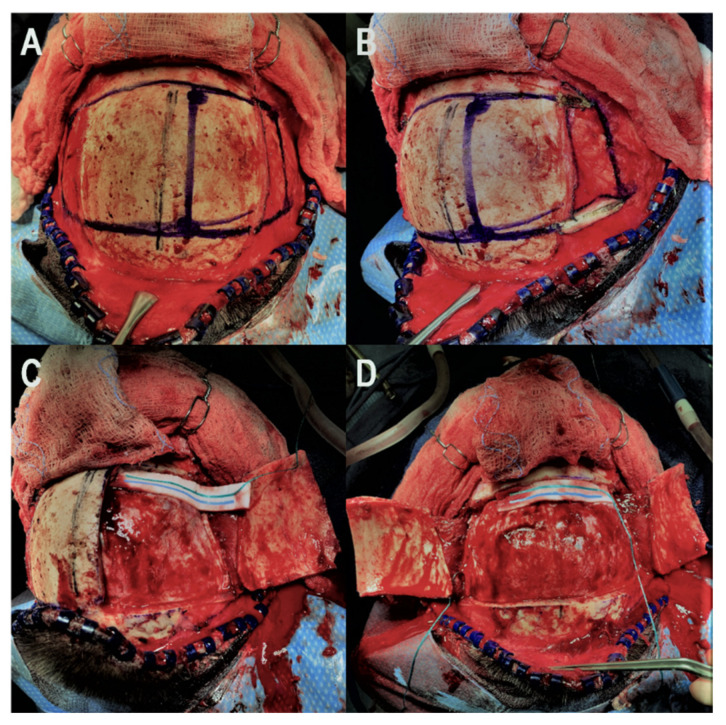
Illustration of the key steps for the bifrontal osteoplastic flap technique. (**A**): The midline is identified using the sagittal suture and the six necessary burr holes are marked. (**B**): Monopolar electrocautery is used to cut the temporalis muscle and expose the keyhole and posterior temporal burr hole on the right side. (**C**): The four right-sided burr holes are drilled and connected to elevate the right-sided osteoplastic flap. (**D**): The superior sagittal sinus is epidurally dissected and step (**C**) is performed on the left side to elevate the contralateral osteoplastic flap.

**Figure 2 brainsci-12-00163-f002:**
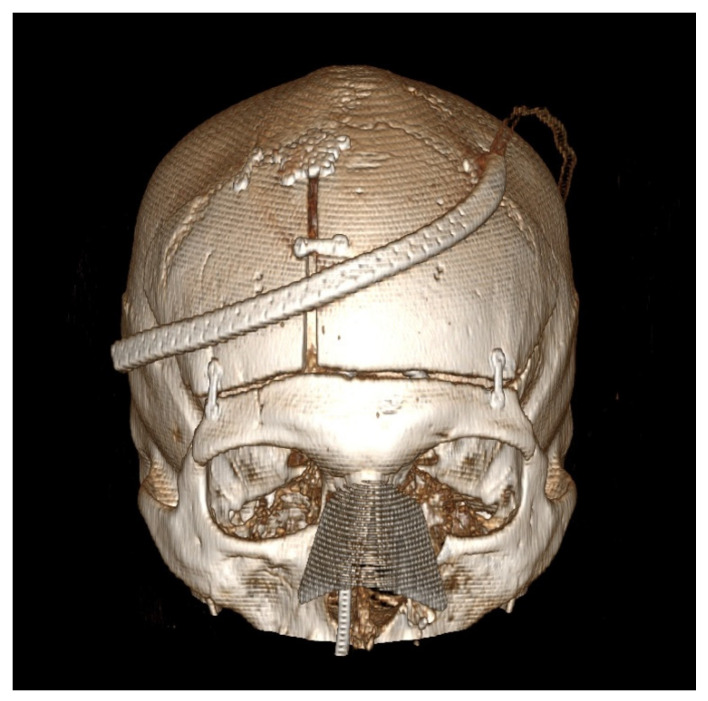
Post-operative computed tomography 3D reconstruction of the patient is presented in Figure 1. This patient underwent a combined bifrontal osteoplastic flap and transnasal approach for resection of a sinonasal adenocarcinoma with extension through the cribriform plate. Note the position of the fixation hardware to ensure a cosmetically pleasing cranioplasty.

**Table 1 brainsci-12-00163-t001:** Patient demographics.

	Infected Cases	Non-Infected Cases	*p* Value
Age (mean)	54.3 ± 5.0	56.3 ± 16.2	0.77
Gender (F, %)	2 (33.3%)	6 (42.9%)	0.7
Diagnosis (%)			
Cribriform defect	0	2 (14.3%)	0.34
Cribriform meningioma	0	2 (14.3%)	0.34
Esthesioneuroblastoma	4 (66.6%)	3 (21.4%)	0.058
Nasal squamous cell carcinoma	1 (16.6%)	2 (14.3%)	0.9
Planum meningioma	0	4 (28.6%)	0.15
Sinonasal adenocarcinoma	1 (16.6%)	1 (7.1%)	0.5
Pre-op chemotherapy	1 (16.6%)	0 (0%)	0.12
Pre-op radiotherapy	1 (16.6%)	0 (0%)	0.12
Skull base repair			
Vascularized pericranial graft	1 (16.6%)	0 (0%)	0.12
Split thickness calvarial graft + vascularized pericranial flap	2 (33.3%)	7 (50%)	0.5
Vascularized pericranial graft + MEDPOR	2 (33.3%)	4 (28.6%)	0.83
Vascularized pericranial graft + titanium mesh	1 (16.6%)	3 (21.4%)	0.81
Duration of surgery in mins (mean)	534.2 ± 77.0	453.6 ± 122.3	0.15
Post-op chemotherapy	5 (83.3%)	4 (28.6%)	**0.02**
Post-op radiotherapy	5 (83.3%)	6 (42.9%)	0.1
Re-operation rate for recurrence (%)	1 (16.6%)	1 (17.1%)	0.98
Follow-up times (months)	87 ± 53.9	91.2 ± 74.5	0.9

**Table 2 brainsci-12-00163-t002:** Types of infection and their management.

Case	Type of Infection	Craniectomy	Organism	Antibiotic Treatment	Time to Cranioplasty	Type of Cranioplasty
1	Epidural abscess and osteomyelitis	Y	MRSA	6 w of IV vancomycin	12 weeks	PEEK implant
2	Osteomyelitis	Y	MSSA	6 w of IV nafcillin	28 weeks	PEEK implant
3	Epidural abscess and osteomyelitis	Y	*S. marcenses*, *S. intermedius*, multiple anaerobic species	6 w of IV meropenem	N/A	N/A
4	Superficial forehead abscess	N	MRSA	4 w of IV vancomycin and PO rifampin, doxycycline chronic suppression	N/A	N/A
5	Epidural abscess, osteomyelitis and meningitis	Y	MSSA, *S. anginosus*	6 w of IV nafcillin	24 weeks	PEEK implant
6	Osteomyelitis	Y	MRSA	6 w of IV vancomycin	24 weeks	PEEK implant

MRSA, methicillin-resistant *S. aureus*; MSSA, methicillin-resistant *S. aureus*; PEEK, polyether ether ketone.

## Data Availability

Data available on request due to restrictions eg privacy or ethical. The data presented in this study are available on request from the corresponding author. The data are not publicly available due to request not included in IRB permission for study.

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
