# Peer review of "Bifrontal Osteoplastic Flap: An Option to Decrease Infection in Bifrontal Craniotomies with Skull Base Osteotomies"

_brainsci, 2022, doi:10.3390/brainsci12020163_

Round 1
Reviewer 1 Report
The Authors report their experience with bifrontal craniotomy with specific interest on infection and propose an alternative technique to improve the outcome of this approach.
In order to understand the risk factors for infection it would be better to display information regarding non-infected patients.
Adjuvant radio and chemotherapy are associated with infections in this case series. When we're adjuvant therapy performed? In my experience radio and chemotherapy are performed at least one month after surgery.
I would like to know if ORL collaboration consisted of a trans nasal approach. Was any nasal flap applied?
Please re-read the paper to correct small language mistakes.
Author Response
The Authors report their experience with bifrontal craniotomy with specific interest on infection and propose an alternative technique to improve the outcome of this approach.
In order to understand the risk factors for infection it would be better to display information regarding non-infected patients.
Data in revised Table 1
Adjuvant radio and chemotherapy are associated with infections in this case series. When we're adjuvant therapy performed? In my experience radio and chemotherapy are performed at least one month after surgery.
Radiation and chemotherapy administrated 6 weeks after surgery, as stated in lines 74-75, with the exception of one patient so received pre-operative radiation, as stated in lines 73-74
I would like to know if ORL collaboration consisted of a trans nasal approach. Was any nasal flap applied?
In the 13 cases with otolaryngology collaboration, transnasal approaches were used, but nasal flaps were not applied to due to the extent of mucosal dissection, as stated in lines 75-79.
Please re-read the paper to correct small language mistakes.
The manuscript was edited for language, as requested
Reviewer 2 Report
The authors present their institutional experience of bifrontal cranitomies with skull base osteotomies, focusing on their infection rates, and propose the use of bifrontal osteoplastic flap for infection prevention. Although they only report two patients treated with this new technique, their work is interesting and should be further investigated also with larger cohorts.
Following I have some comments:
Major comment:
1) Result section: Data on treatment strategies are lacking, especially for the 14 patients without a postoperative infection. If available, I suggest to report rates of preoperative management (if any) (e.g., pre-surgery radiation and/or chemotherapy), surgical approaches, type of skull base repair and type of grafts (also for non-infection cases), duration of surgery, postoperative treatment (rates of chemotherapy and radiotherapy should be distinguished one another), re-operation for tumor recurrence (if any), and follow-up times. A new "Table 1" presenting summarized frequencies and percentages for each variable, divided into infection cohort vs non-infection cohort, may be useful for providing the readers with more information on potential infection risks after skull base surgery.
Minor comments:
2) Lines 57-58: "Craniectomies 58 for traumatic injuries were excluded were excluded." -> There is a repetition of the words "were excluded". Please delete.
3) The noun "data" is plural, please correct this in the entire text.
Author Response
The authors present their institutional experience of bifrontal cranitomies with skull base osteotomies, focusing on their infection rates, and propose the use of bifrontal osteoplastic flap for infection prevention. Although they only report two patients treated with this new technique, their work is interesting and should be further investigated also with larger cohorts.
Following I have some comments:
Major comment:
1) Result section: Data on treatment strategies are lacking, especially for the 14 patients without a postoperative infection. If available, I suggest to report rates of preoperative management (if any) (e.g., pre-surgery radiation and/or chemotherapy), surgical approaches, type of skull base repair and type of grafts (also for non-infection cases), duration of surgery, postoperative treatment (rates of chemotherapy and radiotherapy should be distinguished one another), re-operation for tumor recurrence (if any), and follow-up times. A new "Table 1" presenting summarized frequencies and percentages for each variable, divided into infection cohort vs non-infection cohort, may be useful for providing the readers with more information on potential infection risks after skull base surgery.
These data are included in a revised Table 1.
Minor comments:
2) Lines 57-58: "Craniectomies 58 for traumatic injuries were excluded were excluded." -> There is a repetition of the words "were excluded". Please delete.
The requested deletion was made
3) The noun "data" is plural, please correct this in the entire text.
"Data" is treated as plural throughout the manuscript.
Reviewer 3 Report
Infection in bifrontal craniotomies is an issue of great clinical importance since the infection can cause several complications, including permanent and often irreversible consequences. For this reason, the paper submitted for my assessment is a valuable contribution to the discussion on the choice of the best management option. The article is well-written, and my only reservations are raised by the need for a broader discussion of the current literature.
The following papers are essential to discuss:
Johnson WC, Ravindra VM, Fielder T, Ishaque M, Patterson TT, McGinity MJ, Lacci JV, Grandhi R. Surface Area of Decompressive Craniectomy Predicts Bone Flap Failure after Autologous Cranioplasty: A Radiographic Cohort Study. Neurotrauma Rep. 2021 Aug 27;2(1):391-398. doi: 10.1089/neur.2021.0015.
Kim MJ, Lee HB, Ha SK, Lim DJ, Kim SD. Predictive Factors of Surgical Site Infection Following Cranioplasty: A Study Including 3D Printed Implants. Front Neurol. 2021 Nov 2;12:745575. doi: 10.3389/fneur.2021.
Authors should also include the following sentence: "Following anatomical knowledge during surgical procedures also can reduce the risk of complications, e.g., as a result of tissue traumatization or ischemia." Poblete et al. and should be cited after this sentence:
Poblete, T.; Casanova, D.; Soto, M.; Campero, A.; Mura, J. Microsurgical Anatomy of the Anterior Circulation of the Brain Adjusted to the Neurosurgeon's Daily Practice. Brain Sci. 2021, 11, 519. https://doi.org/10.3390/brainsci11040519
Author Response
Infection in bifrontal craniotomies is an issue of great clinical importance since the infection can cause several complications, including permanent and often irreversible consequences. For this reason, the paper submitted for my assessment is a valuable contribution to the discussion on the choice of the best management option. The article is well-written, and my only reservations are raised by the need for a broader discussion of the current literature.
The following papers are essential to discuss:
Johnson WC, Ravindra VM, Fielder T, Ishaque M, Patterson TT, McGinity MJ, Lacci JV, Grandhi R. Surface Area of Decompressive Craniectomy Predicts Bone Flap Failure after Autologous Cranioplasty: A Radiographic Cohort Study. Neurotrauma Rep. 2021 Aug 27;2(1):391-398. doi: 10.1089/neur.2021.0015.
Kim MJ, Lee HB, Ha SK, Lim DJ, Kim SD. Predictive Factors of Surgical Site Infection Following Cranioplasty: A Study Including 3D Printed Implants. Front Neurol. 2021 Nov 2;12:745575. doi: 10.3389/fneur.2021.
Additional text with reference to these two papers is included in lines 117-120.
Authors should also include the following sentence: "Following anatomical knowledge during surgical procedures also can reduce the risk of complications, e.g., as a result of tissue traumatization or ischemia." Poblete et al. and should be cited after this sentence:
Poblete, T.; Casanova, D.; Soto, M.; Campero, A.; Mura, J. Microsurgical Anatomy of the Anterior Circulation of the Brain Adjusted to the Neurosurgeon's Daily Practice. Brain Sci. 2021, 11, 519. https://doi.org/10.3390/brainsci11040519
This sentence has been added and referenced in lines 120-123.